# Association between Mothers’ Emotional Problems and Autistic Children’s Behavioral Problems: The Moderating Effect of Parenting Style

**DOI:** 10.3390/ijerph20054593

**Published:** 2023-03-05

**Authors:** Xiujin Lin, Lizi Lin, Xin Wang, Xiuhong Li, Muqing Cao, Jin Jing

**Affiliations:** Department of Maternal and Child Health, School of Public Health, Sun Yat-Sen University, Guangzhou 510080, China

**Keywords:** depression, anxiety, parenting style, behavioral problem, autism

## Abstract

Mothers’ emotional problems are associated with autistic children’s behavioral problems. We aim to test whether parenting styles moderate associations between mothers’ mood symptoms and autistic children’s behavioral problems. A sample of 80 mother–autistic child dyads were enrolled at three rehabilitation facilities in Guangzhou, China. The Social Communication Questionnaire (SCQ) and the Strengths and Difficulties Questionnaire (SDQ) were used to collect the autistic symptoms and behavioral problems of the children. Mothers’ depression and anxiety symptoms were measured using the Patient Health Questionnaire 9 (PHQ-9) and the General Anxiety Disorder 7-item (GAD-7) scale, respectively, and parenting styles were measured using the Parental Behavior Inventory (PBI). Our results show that mothers’ anxiety symptoms were negatively associated with their children’s prosocial behavior scores (β = −0.26, *p* < 0.05) but positively related to their social interaction scores (β = 0.31, *p* < 0.05). Supportive/engaged parenting styles positively moderated the effects of mothers’ anxiety symptoms on their prosocial behavior score (β = 0.23, *p* = 0.026), whereas hostile/coercive parenting styles had a negative moderation (β = −0.23, *p* = 0.03). Moreover, hostile/coercive parenting styles positivity moderated the effects of mothers’ anxiety symptoms on social interaction problems (β= 0.24, *p* < 0.05). The findings highlight, where mothers adopted a hostile/coercive parenting style while experiencing high anxiety, their autistic child may have more serious behavioral problems.

## 1. Introduction

Children with autism spectrum disorder (ASD) may exhibit persistent deficits in social communication and social interaction as well as restricted and repetitive patterns of behavior [1]. Regardless of the core symptoms, behavioral problems are also commonly observed but difficult to manage among children with ASD [2]. Thus, parenting children with ASD is challenging [3,4,5] and mood symptoms and disorders are widely reported in parents of autistic children [6,7,8]. According to a previous study, mothers with stable and positive moods are supportive of functional improvements in their children with ASD; thus, providing support for mood and mood-related problems in mothers of children with ASD is beneficial for the child and the mother [9].

We reviewed previous studies regarding parents’ emotional and child behavioral problems, research in the general population has identified well-established connections between parents’ mental health difficulties and the symptoms and behavioral problems of their children [10,11,12]. A number of research studies have reported on the relationship between mothers’ mood symptoms and behavioral problems in children with ASD [13], and those with autistic symptoms [8,14]. Furthermore, children with ASD would have a better prognosis, including social interaction, attention problems, and hyperactivity/inattention symptomatology, when parents had healthy and positive emotional states [9,15,16]. Previous research had limitations, such as a lack of focus on the possible moderating variables that could explain the relationship between anxiety and depression symptoms in mothers and the behavioral problems and symptoms of their children with ASD. Thus, understanding the role of mothers’ depression and anxiety symptoms is important in the development and maintenance of behavioral difficulties in children with ASD.

Research has established that parenting has a critical influence on a child’s development, such as their social development [17] and self-esteem [18], and is also related to the quality of the parent-child relationship [19]. A series of studies has shown that parenting styles are associated with the behavioral problems of children with ASD, such as externalizing problems [20] or internalizing behavioral problems [21]. In addition, parenting styles may affect the intervention and rehabilitation of ASD, for example, a positive parenting style predicts better social competence in children with ASD [22,23]. However, mothers with anxiety symptoms have a more negative parenting style, such as intrusive involvement in anxious children compared to children with typical development (TD) [24]. Hentges et al. found that mothers’ depression has an indirect effect on internalizing problems in children with TD via hostile parenting [25]. Thus, understanding how parenting style interacts with mothers’ mood problems and autistic children’s behavioral problems will help with the development of targeted interventions, as well as understanding whether parenting style may worsen or protect against these effects.

In the current study, we recruited 2–12-year-old autistic children and their mothers to examine the possible moderating influence of parenting style in the association between the mothers’ mood problems and their autistic children’s behavioral problems. Based on existing research, the current study holds several hypotheses: (1) Mothers with higher levels of anxiety and depression will have autistic children with more behavioral problems; (2) Parenting styles will moderate the association between mothers’ moods and behavioral problems in autistic children. More specifically, those with hostile/coercive parenting styles and high levels of anxiety and depression are associated with more behavioral problems in their autistic children. To the best of our knowledge, this study is the first to consider the effects of mothers’ moods and parenting styles on their autistic children’s behavioral problems.

## 2. Materials and Methods

### 2.1. Participants and Procedures

Data were from an ongoing study that started in September 2020 in Guangzhou, China. Children from ages 2 to 12 years were recruited from three special schools affiliated with the Guangzhou Disabled Persons Federation. The inclusion was restricted to children diagnosed with ASD by a child psychiatrist (according to DSM-V). Children with other neurodevelopmental abnormalities, such as epilepsy and cerebral palsy, and physically handicapped were excluded. A total of 110 parent–child (with ASD) dyads agreed to participate in the study. After excluding all questionnaires submitted by fathers, we were left with 80 mother–child dyads for the final analysis. Written informed consent from the mothers had been obtained before the questionnaire and behavior evaluations were completed. The mothers were asked to finish a structured questionnaire (including anxiety, depression symptoms, parenting styles, child behavioral problems, and demographic information, described below), with a uniform guide. Autistic children were assessed for cognition by licensed researchers who had been in standardized training. This study was approved by the Ethics Committee of the School of Public Health at Sun Yat-sen University.

### 2.2. Scales in the Questionnaire

Mothers’ anxiety: The General Anxiety Disorder 7-item (GAD-7) scale, a brief, seven-item self-report scale designed to assess generalized anxiety in mothers was used to evaluate anxiety symptoms [26]. Each of the seven items is scored from 0 (not at all) to 3 (nearly every day). The total GAD-7 scale score ranges from 0 to 21. A higher score indicates greater symptoms on the GAD-7. The current study used the recommended mild-to-severe cut-off scores for anxiety (GAD-7 ≥ 5) to classify subjects with or without a history of anxiety. The Cronbach’s α among Chinese was 0.898 [27].

Mothers’ depression: Symptoms of depression were evaluated by Patient Health Questionnaire 9 (PHQ-9), a nine-item self-report scale designed to assess symptoms of depression in mothers [28]. Each of the nine items can be scored from 0 (not at all) to 3 (nearly every day), and the total scale score ranges from 0 to 27. A higher score indicates greater symptoms on the PHQ-9. The current study used the recommended mild-to-severe cut-off scores for anxiety (PHQ-9 ≥ 5) to classify subjects with or without a history of anxiety. The Cronbach’s α among Chinese was 0.85 [29].

Parenting style: We used the Parental Behavior Inventory (PBI) to evaluate the mothers’ parenting styles. The PBI was designed by Love-Joy [30]; it is a parent’s self-evaluation of their parenting behavior with preschool and junior school children. It is a 20-item self-rated questionnaire including support/participation, and hostility/coercion parenting styles. The Cronbach’s α of support/participation and hostility/coercion were 0.807 and 0.652, respectively in Chinese [31].

Children’s behavioral problems: We used the Strengths and Difficulties Questionnaire (SDQ) to evaluate the behavioral problems of children. Mothers were asked to complete the extended version of the SDQ for children with ASD. The SDQ is a 25-item questionnaire that represents a problem of hyperactivity/inattention (SDQ-HA, 5 items), emotional symptoms (SDQ-ES, 5 items), peer problems (SDQ-PP, 5 items), conduct problems (SDQ-CP, 5 items), and prosocial behavior (SDQ-PB, 5 items); it is designed to assess the behavioral and emotional problems in children and adolescents [32]. Each of the 25 items is rated as being not true (0), somewhat true (1), or certainly true (2), and each of the SDQ subscales consists of five items, thereby yielding scores between 0 and 10. The hyperactivity/inattention, emotional symptoms, peer problems, and conduct problems subscales produce a score for total difficulties, which can range between 0 and 40. A higher score indicates more deficient functioning. For the strength score of the prosocial subscale, a higher score indicates better functioning. The current study used the recommended cut-off scores for each subscale (SDQ-HA ≥ 7, SDQ-ES ≥ 7, SDQ-PB ≤ 4, SDQ-PP ≥ 6, and SDQ-CP ≥ 5) to classify children with ASD as with or without a history of behavioral problems. The SDQ has good reliability and structural validity in Chinese individuals [33].

Autistic behaviors: The Social Communication Questionnaire (SCQ) was used to evaluate the core symptoms of autism. The SCQ scale is a 40-item scale for parents or caregivers designed as a brief screening measure of ASD [34]. The items are based on those with the most discriminative diagnostic efficacy in ADI-R. The SCQ is mainly divided into three areas, namely, the social interaction domain (S), the communication domain (C), and the restricted, repetitive, and stereotyped patterns of behavior domain (R). All items were answered by “Yes” or “No” (0 = no abnormal behavior, 1 = abnormal behavior). A higher score indicates greater symptoms on that subscale. Cronbach’s α coefficient for the present study was 0.89.

#### Demographic Information

Baseline characteristics were recorded using written questionnaires, including the mother’s age, education, ethnicity, age and gender of the child with ASD, family income, and the number of family members.

### 2.3. Statistical Analysis

The study was designed to answer another question, but the collected data were used here to address the current questions. SPSS v23.0 statistical software was used to conduct statistical analysis. The descriptive statistics for continuous variables were presented as the mean (*M*) and standard deviation (*SD*), and the count data were described by prevalence (%). Pearson’s correlations were conducted on maternal anxiety and depression symptoms, parenting style, and behavioral problems in children with ASD. Multiple linear regression was used to determine if parenting style moderated the associations between mothers’ anxiety or depression symptoms and behavioral problems in children with ASD. We added mothers’ anxiety or depression symptoms and the parenting style in the first step, and the interaction of mothers’ anxiety or depression symptoms and the parenting style in the second step. All regression analyses included children’s gender and age, family income, and maternal education as covariates in the third step. Standardized regression coefficients presented all betas, and the significant level was *p* < 0.05. A simple slope analysis was conducted using the Process 2.16 macro plug-in of SPSS 23.0.

## 3. Results

### 3.1. Demographic Information

Table 1 outlines the sample demographic information for the children included in the analysis. In children with ASD (87.5% boys), 53 (66.3%) children were under 6 years old. As for the mothers, 69.3% were more than 35 years old, 72.5% have low education (less than 9 years), and 76.3% have a family income of less than 8000 yuan per month.

### 3.2. Prevalence of Behavioral Problems in Children with Autism and Mothers’ Emotional Problems

The prevalence of abnormal SDQ-HA, SDQ-ES, SDQ-PB, SDQ-PP, and SDQ-CP behavioral problems (shown in Appendix A) among children with ASD was 53 (66.3%), 5 (6.3%), 62 (77.5%), 71 (88.8%), and 13 (16.3%), respectively. In addition, the prevalence of depression and anxiety symptoms in mothers with children with ASD was 31 (38.8%) and 38 (37.5%), respectively. The mean and standard deviation of the score of the supportive/engaged parenting style was 33.03 ± 7.74, and that of the hostile/coercive parenting style was 17.91 ± 6.49.

### 3.3. Correlation Analysis

In the correlation analysis between the mothers’ mood symptoms and children’s symptoms (Table 2), the mothers’ depression along with anxiety symptoms were positively associated with the children’s hyperactivity score (*r* = 0.28, 0.29 for depression and anxiety, respectively; *p* < 0.05) and negatively associated with the prosocial behavior score (*r* = −0.27 −0.26 for depression and anxiety, respectively; *p* < 0.05). In addition, mothers’ depression symptom was related to a higher conduct problems score (*r* = 0.25, *p* < 0.05). As for the parenting style, supportive/engaged was associated with lower SCQ scores in social (*r* = −0.31, *p* < 0.05), repetitive (*r* = −0.33, *p* < 0.05), and SCQ total score (*r* = −0.33, *p* < 0.05); however, hostile/coercive was associated with a higher score in the SDQ total score (*r* = 0.24, *p* < 0.05) and communicating domain (*r* = 0.24, *p* < 0.05).

### 3.4. Relationship between Mothers’ Anxiety Symptoms and Children’s Prosocial Behaviors Measured by SDQ Moderated by Parenting Style

After adjusting for the children’s gender and age, family income, and mothers’ education, multiple linear regression analysis showed that mothers’ anxiety symptoms were negatively associated with children’s prosocial behavior (β = −0.26, *p* < 0.05); and supportive/engaged parenting style had a marginally positive relationship to children’s prosocial behavior (β = 0.21, *p* = 0.051). In addition, a supportive/engaged parenting style positively moderated the effect of mothers’ anxiety symptoms on children’s prosocial behavior (β = 0.23, *p* = 0.026); conversely, a hostile/coercive parenting style negatively moderated the effect of mothers’ anxiety on children’s prosocial behavior (β = −0.23, *p* = 0.031, see Table 3). The negative results of mothers’ depression and children’s behavioral problems collected in the SDQ are shown in Appendix A.

We used a simple slope analysis taking into account the effect of the interaction between mothers’ anxiety symptoms and supportive/engaged parenting style on children’s prosocial behavior. As an example, we set the two special values of mothers’ anxiety symptoms and supportive/engaged parenting style as a standard deviation above and below the average. In addition, we calculated the simple slope of maternal anxiety symptoms on children’s prosocial behavior when a supportive/engaged parenting style was high/low. The results showed that mothers’ anxiety symptoms were negatively associated with children’s prosocial behavior when the supportive/engaged domain was low (*b* = −0.282, *p* < 0.01), whereas there was no significant association when the supportive/engaged domain was high (*b* = −0.212, *p* > 0.05, see Figure 1a).

For high levels of hostile/coercive parenting styles, mothers’ anxiety symptoms were negatively associated with children’s prosocial behavior (*b* = −0.300, *p* < 0.01), but had no association when the hostile/coercive domain was low (*b* = −0.063, *p* > 0.05, see Figure 1b).

### 3.5. Effects of Mothers’ Anxiety Symptoms on Children’s Social Interaction Moderated by Parenting Style and Measured by SCQ

After adjusting for children’s gender and age, family income, and mothers’ education, multiple linear regression analysis showed that hostile/coercive parenting styles positively moderated the effect of mothers’ depression on children’s social interaction (β = 0.24 for hostile/coercive; *p* < 0.05; see Table 4). Supportive/engaged marginally moderated the effect of mothers’ depression on children’s social interaction (β = 0.20; *p* = 0.052; see Table 4). The results of mothers’ depression and children’s other behavioral problems by SCQ are shown in Appendix A.

The simple slope showed that mothers’ anxiety symptoms were positively associated with children’s social interaction with a high hostile/coercive parenting style (*b* = 0.423, *p* < 0.01) but not with a low hostile/coercive parenting style (*b* = 0.114, *p* > 0.05, see Figure 2).

## 4. Discussion

Parenting a child with autism is difficult; emotional problems, including depression and anxiety, have been widely reported among parents of autistic children [8,35]. By using a cross-sectional study, we tested if parenting styles moderated the association between mothers’ emotional symptoms and autistic children’s behavioral problems and social communication. The main findings of this study confirmed that (1) mothers’ anxiety and depression symptoms are positively associated with the severity of behavioral problems among children with ASD; and (2) moreover, when parenting style is low supportive/engaged, or high hostile/coercive, mothers’ anxiety symptoms are associated with a decrease in children’s prosocial behavior or an increase in social interaction problems.

Our findings indicated that mothers’ anxiety was associated with less prosocial behavior or more social interaction problems in autistic children [8,14]. We proposed several mechanisms to explain why a negative parenting style may worsen the situation. (1) More anxiety symptoms can cause mothers to adopt negative parenting strategies, such as aggressive behavior and violence [36], thereby eroding children’s self-esteem and their ability to regulate their own emotions [37] and increasing children’s behavioral problems. On the contrary, a supportive parenting style will provide children with a positive parent–child interaction, which the child may use when interacting with others, thereby benefitting their peer relationships and social belonging, having enough social support when encountering problems, and decreasing the rate of problem behaviors [38,39,40,41]. (2) Research on children with TD showed that negative parenting behaviors, such as being hostile/coercive, can function as a risk factor during children’s behavioral problem development; whereas positive parenting behaviors can be a protective factor [25,42]; children with ASD may share the same mechanism. (3) Mothers’ anxiety symptoms lead to intrusive, hostile, and neglectful behaviors, as well as less involvement in parenting their children [43,44], which may cause problematic parent–child interactions, thereby finally decreasing the prosocial behaviors of children with ASD [45].

A negative correlation between mothers’ depression symptoms and children’s prosocial behavior [46,47] is found among TD, which is similar to our findings in autistic children. This finding may be attributed to the incapability of depressed mothers to respond to their children’s needs, which limits the whole family’s initiative to seek proper intervention for the children [48]. Conversely, mothers’ internal physiological changes can be inherited by their children, which can increase the emotional and behavioral problems of autistic children [49]. However, the relationship between mothers’ depression symptoms and children’s behavioral problems is not moderated by parenting style in the current study, which can be attributed to the limited sample size.

The present study found that mothers of children with ASD had a comparable rate of depression and anxiety symptoms (38.8% and 37.5%, respectively) with previous research [50,51,52], which confirmed the elevated risk of depression among mothers of autistic children. We also compared our rate with the rate from other regions of China [8,53]. We found that anxiety and depression symptoms among mothers with children with ASD were more prevalent in the present study. In addition, our study addressed the most salient problem behaviors in autistic children, which were hyperactivity/inattention, prosocial behavior, and peer problems; this was similar to the previous research [54,55]. Our current findings corroborate and further attest that children with ASD often exhibit co-occurring behavioral problems. According to Coplan et al., pragmatic language is a possible pathway in the development of behavioral problems, as it plays an important role in children’s communication with peers, especially in the school-age period. Children solve problems and achieve social goals through adequate language skills, which, for autistic children, are lacking [56]. Thus, limited language skills may make them feel insecure about engaging in peer relationships and cause more problematic behaviors [57]. Moreover, previous studies suggested that other features of autism such as sensory difficulties [58] or resistance to change [59] also caused more problem behaviors.

Our results also confirmed that a hostile/coercive parenting style was positively related to hyperactivity and the total behavioral problem level of autistic children (see Appendix A), which was reported by Maljaars et al. [21]. The coercion theory [42] proposes that in a coercive cycle, aversive child behaviors reciprocally influence parenting behaviors, which results in the negative reinforcement of undesirable behaviors in children and parents [60]. Conversely, a positive parenting style is associated with the prosocial behaviors of children with TD [61,62]. This finding is in line with ours (see Appendix A), which also supported the coercion theory.

This study has a few limitations. First, in view of the cross-sectional design, the causal relationship was not addressed, and the negative finding of an interaction effect between parenting style and depressive mood could be attributed to the relatively small sample size. We recommend a longitudinal study with a larger sample size to clarify the potential mechanism of the association. Second, our study did not consider the role of fathers; the parenting style and emotional symptoms of fathers should be considered in future studies. Third, as there was no normative or clinical comparison group, a more robust design should be considered. Fourth, a limitation of our study was that we used maternal reports of children’s behavioral problems with a single informant (i.e., the mother). Future studies should include multiple informants, such as teachers and fathers to further explore these associations.

## 5. Conclusions

In this research, we confirmed a high rate of anxiety and depression symptoms among autistic children’s mothers, as well as behavioral problems of autistic children. High levels of anxiety and depression in mothers are linked to more behavioral problems in their autistic children. Negative parenting styles, such as low supportive/engaged or high hostile/coercive, further enhance the association between mothers’ mood problems and less prosocial behaviors, and more serious social interaction problems among these children. So far most parenting programs aimed at parents of children with ASD have focused on improving communication in children; studies addressing parenting strategies are limited [63]. Thus, we propose that parents may need more support in coping with emotional problems and improving their parenting skills to decrease the problem behavior of autistic children.

## Figures and Tables

**Figure 1 ijerph-20-04593-f001:**
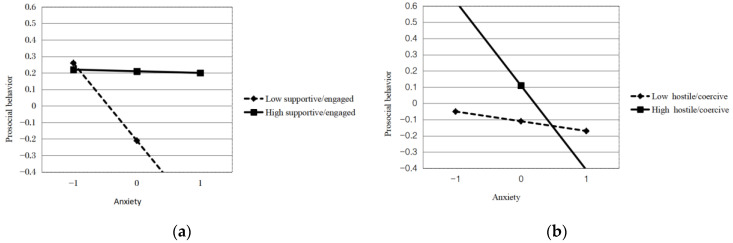
(**a**) Supportive/engaged parenting style moderated the associations between mothers’ anxiety and prosocial behavior; (**b**) Hostile/coercive parenting style moderated the associations between mothers’ anxiety and prosocial behavior.

**Figure 2 ijerph-20-04593-f002:**
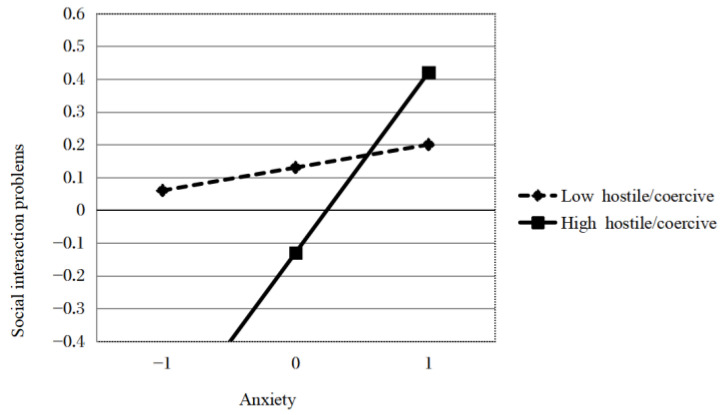
Hostile/coercive parenting style moderated the associations between mothers’ anxiety and social interaction problems.

**Table 1 ijerph-20-04593-t001:** Basic characteristics of samples.

Variables		n	%	
Gender	Boys	70	87.5	
	Girls	10	12.5	
Age				
	<6 years	53	66.3	
	≥6 years	27	33.8	
Maternal age				
	<35 years	31	30.7	
	≥35 years	70	69.3	
Maternal education				
	Low (primary, secondary, high school, and uneducated)	58	72.5	
	High (university and above)	22	27.5	
Family income				
	<8000	61	76.3	
	≥8000	19	23.8	

**Table 2 ijerph-20-04593-t002:** Mean value, standard deviation, and correlation coefficient of each variable (n = 80).

Variable	Depression	Anxiety	Supportive/Engaged	Hostile/Coercive	M ± SD
SDQ-HA	0.28 *	0.29 **	−0.19	0.06	7.10 (1.80)
SDQ-ES	−0.08	−0.07	0.05	0.10	2.65 (1.89)
SDQ-PB	−0.27 *	−0.26 *	0.19	0.12	2.59 (2.32)
SDQ-PP	0.04	0.00	−0.03	0.08	5.48 (1.831)
SDQ-CP	0.25 *	0.16	−0.19	0.13	2.32 (1.47)
SDQ total scores	0.06	0.03	−0.05	0.24 *	20.148 (4.32)
SCQ-Social interaction	0.08	0.18	−0.31 **	−0.10	6.28 (3.269)
SCQ-Repetitive	0.09	0.16	−0.32 **	0.07	4.30 (2.05)
SCQ-Communicate	0.04	0.11	−0.02	0.24 *	6.18 (3.47)
SCQ-Total	0.08	0.18	−0.33 **	0.02	17.67 (6.68)
M ± SD	4.80 ± 4.58	4.20 ± 3.95	33.03 ± 7.74	17.91 ± 6.49	-

Note: * *p* ≤ 0.05, ** *p* ≤ 0.01. SDQ-HA: hyperactivity/inattention; SDQ-ES: emotional symptoms; SDQ-PB: children’s prosocial behavior; SDQ-PP: peer problems; SDQ-CB: conduct problems; SDQ-TS: total scores of SDQ; SCQ: Social Communication Questionnaire.

**Table 3 ijerph-20-04593-t003:** Hierarchical Linear Regressions of the predictors of SDQ-PB.

	Supportive/Engaged Parenting Style to SDQ-PB			
	Adjusted *β*	*t*	*R* ^2^	*F*
Step 1			0.22	5.38 *
Anxiety	−0.26	−2.54 *		
Supportive/engaged	0.14	1.36		
Step 2			0.26	5.60 *
Anxiety	−0.24	−2.33 *		
Supportive/engaged	0.21	1.98 *		
Supportive/engaged × Anxiety	0.23	2.27 *		
	Hostile/Coercive Parenting Style to SDQ-PB			
	Adjusted *β*	*t*	*R* ^2^	*F*
Step 1			0.18	4.53 *
Anxiety	−0.28	−2.68 **		
Hostile/coercive	0.04	0.36		
Step2			0.23	4.84 *
Anxiety	−0.29	−2.78 **		
Hostile/coercive	0.10	0.90		
Hostile/coercive × Anxiety	−0.23	−2.20 *		

Note: * *p* ≤ 0.05, ** *p* ≤ 0.01. Only significant moderation results are presented. Adjusted variables: children’s gender and age, family income, and mothers’ education; SDQ-PB: child prosocial behavior.

**Table 4 ijerph-20-04593-t004:** Hierarchical Linear Regressions of the predictors of SCQ-Social interaction.

	Supportive/Engaged Parenting Style to SCQ-Social Interaction			
	Adjusted *β*	*t*	*R* ^2^	*F*
Step 1			0.23	5.77 *
Anxiety	0.28	2.66 *		
Supportive/engaged	−0.23	−2.26 *		
Step 2			0.26	5.64 *
Anxiety	0.30	2.92 *		
Supportive/engaged	−0.17	−1.66		
Supportive/engaged × Anxiety	0.20	1.97		
	Hostile/Coercive Parenting Style to SCQ-Social Interaction			
	Adjusted *β*	*t*	*R* ^2^	*F*
Step 1			0.18	4.53 *
Anxiety	0.31	2.87 **		
Hostile/coercive	−0.06	−0.58		
Step2			0.23	4.84 *
Anxiety	0.31	3.00 **		
Hostile/coercive	−0.13	−1.14		
Hostile/coercive × Anxiety	0.24	2.26 *		

Note: * *p* ≤0.05, ** *p* ≤ 0.01. Only significant moderation results are presented. Adjusted variables: children’s gender and age, family income, and mothers’ education; SCQ: Social Communication Questionnaire.

## Data Availability

The data presented in this study are available on request from the corresponding or first author.

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
