# Peer review of "Association between Mothers’ Emotional Problems and Autistic Children’s Behavioral Problems: The Moderating Effect of Parenting Style"

_ijerph, 2023, doi:10.3390/ijerph20054593_

Round 1

Reviewer 1 Report

I found this article very interesting. There are some minor areas that can be improved.

Introduction

There are some repetitions in the introduction.

e.g. "Thus, parenting children with ASD is challenging, and mood symptom, even mood disorders, is widely reported among parents of autistic children [2-4]" AND "Parenting children with ASD is a difficult and taxing experience [6-8]; thus, psychological distress, such as depressive and anxiety symptoms, is common among these parents [3,4]". These sentences are repetitive and can be combined.

Methods

The authors state that children with "a serious sensory disorder" were excluded. a lot of children with ASD have some degree of sensory issues. So how did the authors determine "serious sensory disorder?

The authors state that "A total of 110 parent–child (with ASD) dyads agreed to participate in the study. After data cleaning, we were left 80 with mother–child dyads for the final analysis". It is not clear what is meant by data cleaning? 

Results

I feel that the socio-demographic characteristics of the participants are better included in the results section.

Discussion

In line 280 of the discussion, the authors state that "Mothers of children with ASD suffer from poor mental health". This sentence is repetitive and is better removed.

Authors state that "Children solve problems and achieve social goals by adequate language skills, which, for autistic children, are lacking. Thus, limited language skills may make them feel insecure about engaging in peer relationship and cause more problem behaviors [54]" There are many other reasons that children with ASD have behavioural problems. e.g. sensory difficulties, resistance to change etc.. these should be mentioned. 

Conclusions

The authors state that "Thus, we propose that parents need more support in coping with emotional problems and improving their parenting skills to decrease the problem behavior of autistic children". It might be good to add that, so far most parenting programmes aimed at parents of children with ASD have focused on improving communication in children and studies addressing parenting strategies are limited. See Brookman-Frazee L, Stahmer A, Baker-Ericzén MJ, Tsai K. Parenting interventions for children with autism spectrum and disruptive behavior disorders: opportunities for cross-fertilization. Clin Child Fam Psychol Rev. 2006 Dec;9(3-4):181-200. doi: 10.1007/s10567-006-0010-4. PMID: 17053963; PMCID: PMC3510783.

Reviewer 2 Report

The paper reports a cross-sectional single informant correlational study conducted in a convenience sample of children with autism and their mothers recruited from three special schools in Guangzhou, China. Maternal emotional problems were positively correlated with disruptive behaviours in the children, and negatively correlated with prosocial behaviours. Maternal emotional problems did not correlate with core autistic symptoms in the child. Maternal supportive/engaging parenting behaviours correlated negatively with social interaction problems and repetitive behaviours in the child. Maternal hostile/coercive parenting correlated positively with child communication difficulties. Maternal parenting style did not correlate with child emotional or behavioural problems. Other associations examined were also statistically non-significant.  Multivariate regression modelling showed the relationship between maternal anxiety and child prosocial behaviour was mediated by high supportive/engaging behaviour and low hostile/coercive behaviour. A similar pattern was observed for the association between anxiety and child social interaction problems. The authors concluded that parents of children with autism need more support in coping with their emotional problems and to improve their parenting skills to decrease problem behaviour in their children.

Specific comments         

1.       Single informant studies tend to inflate the apparent association between variables owing to a halo effect. A more robust design would have been maternal report of parent variables, and teacher report of child variables.

2.       It is not clear if the sample was recruited specifically for the present study, or for other purposes. This must be declared.

3.       Association is not causality. The authors acknowledge this, but then proceed to make recommendations that assume causality has been demonstrated.

4.       The cross-sectional design of the study means that the direction of effect is unknown. It is equally plausible that child behaviour difficulties lead to maternal emotional problems and dysfunctional parenting.

5.       TD is mentioned several times in the paper, but never defined. Relevance of the TD studies to the present paper is not clear.  

6.       There were 48 comparisons, therefore the authors should have applied a Bonferroni correction to the threshold for statistical significance.

7.       As there was no normative or clinical comparison group we do not know if the findings are specific to ASD. A more robust design would have been to have a normative or clinical control arm.

8.       Aberrant Behavior Checklist would have been a better instrument than the SDQ for a preschool population. The SDQ is validated for children 4 and above. The sample included 2 and 3 yr olds

9.       The study does not consider the role of fathers, either their impact through direct interaction with the child, or indirect through their interaction with mother

10.   The legend to Table 2 is incomplete (does not define SCQ). Also the Table does not report M and SD for the SDQ and SCQ variables. I wonder in the formatting, whether a column has been left out

11.   The authors do not seem curious why maternal emotion measures correlated exclusively with SDQ subscales while parenting style measures correlated exclusively with SCQ subscales.  

Reviewer 3 Report

It is an interesting study addressing the association between mothers’ emotional problems and autistic children’s behavioral problems and the moderating effects of the association. The authors need to address some comments to improve the manuscript.

Introduction

l  In line 30, explain ‘core symptoms’

l  I personally don’t understand why the authors provide supplementary eTables. I recommend the authors to add the contents of the eTables into the main texts of the manuscripts.

l  Lines 47-53 can be a separate paragraph. Plus in lines 47-49, the authors briefly present the limitations of the previous studies. The description is too brief to persuasively present the need of the study. The authors need to develop literature review more.

l  The paragraph in lines 54-62 can be moved before the description of the limitations of the previous studies.

l  The authors do not sufficiently present the explanations of each variable of the current study in the Introduction.

l  Lines 63-67 are not suitable for the Introduction. It may be moved to the materials and methods section. The hypotheses and the purpose of the study needs to be presented together.

l  Overall, the introduction doesn’t provide sufficient information to persuasively present the need of the study. Introduction needs to be reconstructed and to have more profound literature review to present research gap for presenting the need of the current study. In addition, there are covariates in the analysis, and the authors need to present how the covariates are associated with the variables of the current study in the Introduction so that the covariates need to be controlled.

Method

l  Reformat Table 1

l  To examine the moderating effects, the sample size is quite small. The authors need to provide empirical supports (e.g., G-power) showing that the sample size is OK.

l  In lines 142-145, the demographic information is presented. The information needs to be relocated.

l  Check statistical symbols in Table 2, 3, & 4.

l  In Table 2, the presentation of the correlation values is unclear. Reorganize them by referencing other published papers.

l  In table 3 & 4, the presentation of the multiple regression analyses is unclear. Reorganize them by referencing other published papers.

l  Multicollinearity issues must be checked for multiple regression anaysis

Discussion

l  Write implications, related to covariates.

l  Lines 275-276 are unclear.

l  What are the implications of the lines 280-282.

l  Write practical implications of the current study.

l  Extend limitations.

Round 2

Reviewer 2 Report

The authors have gone some way to addressing the weaknesses in the paper that I thought were modifiable. Thank you for clarifying that the SDQ has been validated for two and three year olds. The discussion about direction of effect still requires some attention. In ADHD, another neurodevelopmental disorder, treatment of child symptoms has been shown to be followed by a spontaneous improvement in parenting. It is quite plausible that the same could occur with ASD. I am not satisfied with the justification for not applying a Bonferroni correction to the level of significance. Revisions to the paper cannot rectify the significant methodological limitations of the study. 

Author Response

Thank you for your patience good comment. I learned a lot through the revision of this study.

Reviewer 3 Report

The authors successfully addressed all the comments. Thank you for their good works.

Author Response

 Thank you for your good comment. I learned a lot through the revision of this study.